# Large and tunable magnetoresistance in van der Waals ferromagnet/semiconductor junctions

Wenkai Zhu [1,2,9], Yingmei Zhu[3,9], Tong Zhou [4], Xianpeng Zhang[5], Hailong Lin[1,2], Qirui Cui[3], Faguang Yan[1], Ziao Wang[1,2], Yongcheng Deng [1], Hongxin Yang[3] ✉, Lixia Zhao[1,6] ✉, Igor Žutić [4] ✉, Kirill D. Belashchenko [7] ✉ & Kaiyou Wang [1,2,8] ✉

Magnetic tunnel junctions (MTJs) with conventional bulk ferromagnets separated by a nonmagnetic insulating layer are key building blocks in spintronics for magnetic sensors and memory. A radically different approach of using atomically-thin van der Waals (vdW) materials in MTJs is expected to boost their figure of merit, the tunneling magnetoresistance (TMR), while relaxing the lattice-matching requirements from the epitaxial growth and supporting high-quality integration of dissimilar materials with atomically-sharp interfaces. We report TMR up to 192% at 10 K in all-vdW $Fe_3GeTe_2/GaSe/Fe_3GeTe_2$ MTJs. Remarkably, instead of the usual insulating spacer, this large TMR is realized with a vdW semiconductor GaSe. Integration of semiconductors into the MTJs offers energy-band-tunability, bias dependence, magnetic proximity effects, and spin-dependent optical-selection rules. We demonstrate that not only the magnitude of the TMR is tuned by the semiconductor thickness but also the TMR sign can be reversed by varying the bias voltages, enabling modulation of highly spin-polarized carriers in vdW semiconductors.

The traditional path to enhance the TMR[1,2] relies on carefully choosing insulators and common ferromagnets, such as MgO with Fe and Co[3,4]. As the MTJ size scales down, this approach poses many obstacles, from materials nonuniformity and deteriorating quality to enhanced energy consumption and reduced stability[5]. The breakthroughs in vdW materials and the discovery of two-dimensional (2D) ferromagnets[6,7] suggest important opportunities to overcome these problems in all-vdW MTJs, where realizing a large TMR ~200% could revolutionize magnetic random-access memories (MRAM)[5].

MTJs with both conventional or vdW ferromagnets typically include an insulating spacer layer instead of the semiconducting barrier layer, owing to an extensive research on insulators such as $Al_2O_3$, MgO, and hBN. However, a realization of tunable spin-polarized transport in semiconductors is desirable for many emerging applications[1]. Conventional materials, such as δ-doped Fe/GaAs junctions, already provide a degree of tunability with bias-dependent sign reversal of interfacial spin polarization and TMR[8–12]. Because the observed spin-dependent signals in such systems are only modest, switching to 2D

[1]State Key Laboratory of Superlattices and Microstructures, Institute of Semiconductors, Chinese Academy of Sciences, 100083 Beijing, China. [2]Center of Materials Science and Optoelectronics Engineering, University of Chinese Academy of Sciences, 100049 Beijing, China. [3]National Laboratory of Solid State Microstructures, School of Physics, Collaborative Innovation Center of Advanced Microstructures, Nanjing University, 210093 Nanjing, China. [4]Department of Physics, University at Buffalo, State University of New York, Buffalo, NY 14260, USA. [5]Department of Physics, University of Basel, Basel, Basel-Stadt CH-4056, Switzerland. [6]Tiangong University, 300387 Tianjin, China. [7]Department of Physics and Astronomy, Nebraska Center for Materials and Nanoscience, University of Nebraska-Lincoln, Lincoln, NE 68588, USA. [8]Beijing Academy of Quantum Information Sciences, 100193 Beijing, China. [9]These authors contributed equally: Wenkai Zhu, Yingmei Zhu. ✉e-mail: hongxin.yang@nju.edu.cn; lxzhao@tiangong.edu.cn; zigor@buffalo.edu; belashchenko@unl.edu; kywang@semi.ac.cn

vdW materials could offer significant advantages by: (i) simultaneously increasing the TMR[13–15] and supporting highly spin-polarized carriers, and (ii) expanding the tunability of spin-dependent properties, as demonstrated, for example, through barrier-thickness controlled spin polarization[16,17] and gate-tunable magnetic proximity effects in hybridized 2D material/ferromagnet interface for spin valves[18–23], and gate-tunable spin galvanic effect in van der Waals heterostructures of graphene with a semimetal or topological insulator[24,25].

Recently, the all-vdW $Fe_3GeTe_2$(FGT)-based MTJs have been widely studied, among which a large TMR of ~300% (4.2 K), ~50% (10 K), and ~110% (4.2 K) have been observed in devices with insulating spacer hBN and devices with semiconductor InSe and $WSe_2$, respectively[26–28]. Compared with an insulator, the advantage of a semiconductor tunnel barrier is that its Fermi level ($E_F$) can be adjusted by doping to make the $E_F$ close to the valence band or closer to the conduction band, which plays an important role in enhancing spin-filtering effect[13]. In addition, a large room-temperature TMR of 85% was observed in $Fe_3GaTe_2$/ $WSe_2$/$Fe_3GaTe_2$ MTJs[29], which confirms the great potential for semiconductor-based MTJs. 2D gallium selenide (GaSe) crystal is a typical layered metal monochalcogenide with an indirect bandgap energy of ~2 eV in the bulk[30], which can serve as a perfect tunnel barrier[31]. Furthermore, it was predicted that the giant magnetoresistance can be obtained by using the semiconductor barrier due to the spin-filtering effect[13,15]. However, the magnetoresistance properties of all-vdW MTJs with GaSe barriers have not been reported yet.

In this work, we not only find that the magnitude of TMR increases first and then decreases with increasing the thickness of the semiconductor spacer GaSe in $Fe_3GeTe_2$/GaSe/$Fe_3GeTe_2$ MTJs, but also find the magnitude and sign of the TMR can be tuned by the bias voltage. The maximum TMR of up to 192% is obtained with 9 layers of a GaSe spacer. This realization greatly expands materials design opportunities for semiconductor spintronics that are unavailable to MTJs with insulators[32,33], including applications in artificial neural networks[34,35] and spin lasers[36].

## Results

The typical optical image of the core structure of the MTJ devices is plotted in Fig. 1a, where the two FGT electrodes are separated by a GaSe layer. To avoid oxidation, we covered the core structure of the device with a hBN flake. The schematic diagram of the device and magnetotransport setup is shown in the inset of Fig. 1a, where an out-of-plane magnetic field **B** controls the magnetization alignment of the FGT electrodes. Our previous works confirmed that GaSe and FGT have high-quality crystal structure[28,37–40]. The photoluminescence spectrum measurement shows that GaSe has a bandgap of ~2 eV (Supplementary Fig. 1). The MTJ devices (A, B, C, D, E, F, and G) with different GaSe-layer thicknesses were fabricated using mechanical exfoliation and dry transfer method (see "Methods"), where the GaSe-layer thicknesses were determined by atomic force microscope (AFM) for device A, B, C, D, E, F, and G are about 5.5, 6.5, 7.3, 8.2, 9.2, 10.0, and 15.6 nm, respectively (Supplementary Fig. 2). To have different coercive fields for the top and bottom electrodes, we select different thicknesses of FGT flakes[28,39,40]. The typical thickness of the bottom and top FGT is ~8 nm and ~12 nm, respectively. From the optical image of all the fabricated devices, the active junction overlap areas $A < 20$ μm², which are comparable to the typical magnetic domain sizes in FGT flakes[41].

We first investigate the current–voltage ($I$–$V_{bias}$) characteristics under applied perpendicular $B = -0.4$ T to ensure the parallel-magnetization configuration of the two FGT. To directly compare different devices, the normalized nonlinear current density-voltage $J$–$V_{bias}$ curves at 10 K for devices A to G are shown in Fig. 1b, where the nonlinear behavior of devices A and B with a thinner GaSe layer are

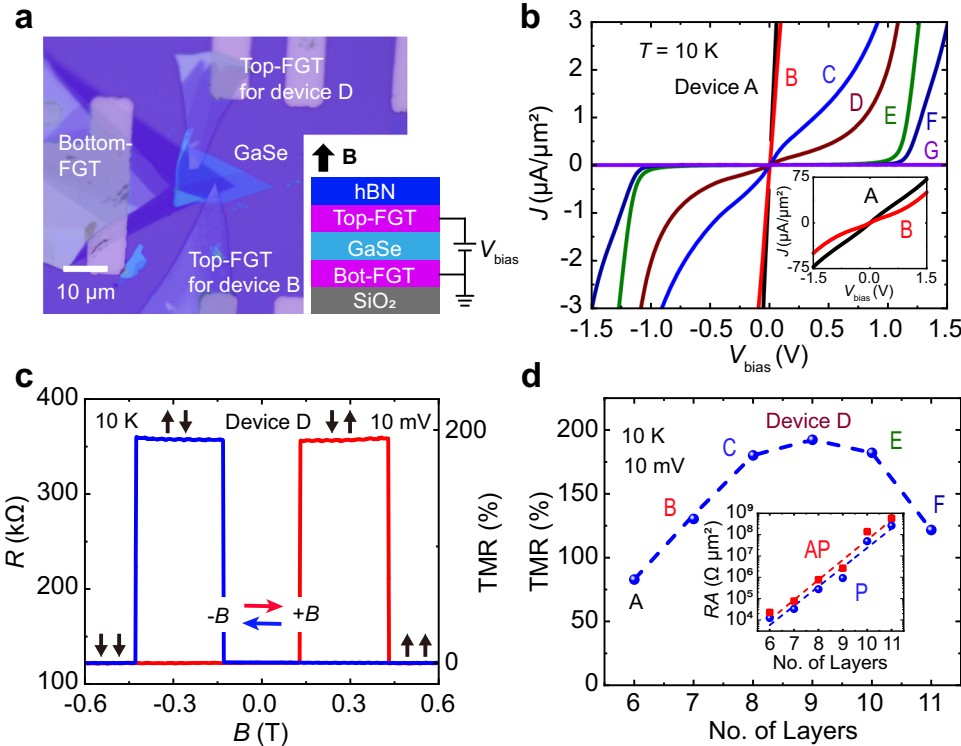

**Fig. 1 | Large TMR in the FGT/GaSe/FGT MTJ devices. a** Typical optical image of the core structure of the device made of different flakes. Inset shows the schematic diagram of the device and magnetotransport setup. The magnetic field (**B**) is applied in an out-of-plane direction. **b** Current density $J$ versus applied bias $V_{bias}$ for the devices with the GaSe thickness ranging from 5.5 to 15.6 nm (devices A-G) in parallel-magnetic configuration. The inset shows the $J$–$V_{bias}$ curves of devices A and B in a larger bias range. **c** Magnetic hysteresis of the resistance $R$ loop for device D at $V_{bias} = 10$ mV, and the corresponding TMR is ~192.4%. Red and blue horizontal arrows show the sweeping directions of **B**. Black-vertical arrows denote the two FGTs' magnetization configurations. **d** The measured maximum TMR ratios in the different devices at 10 mV. The inset shows the plotted zero-bias log($RA$) is nearly linear with the number of GaSe layers in both parallel and antiparallel states.

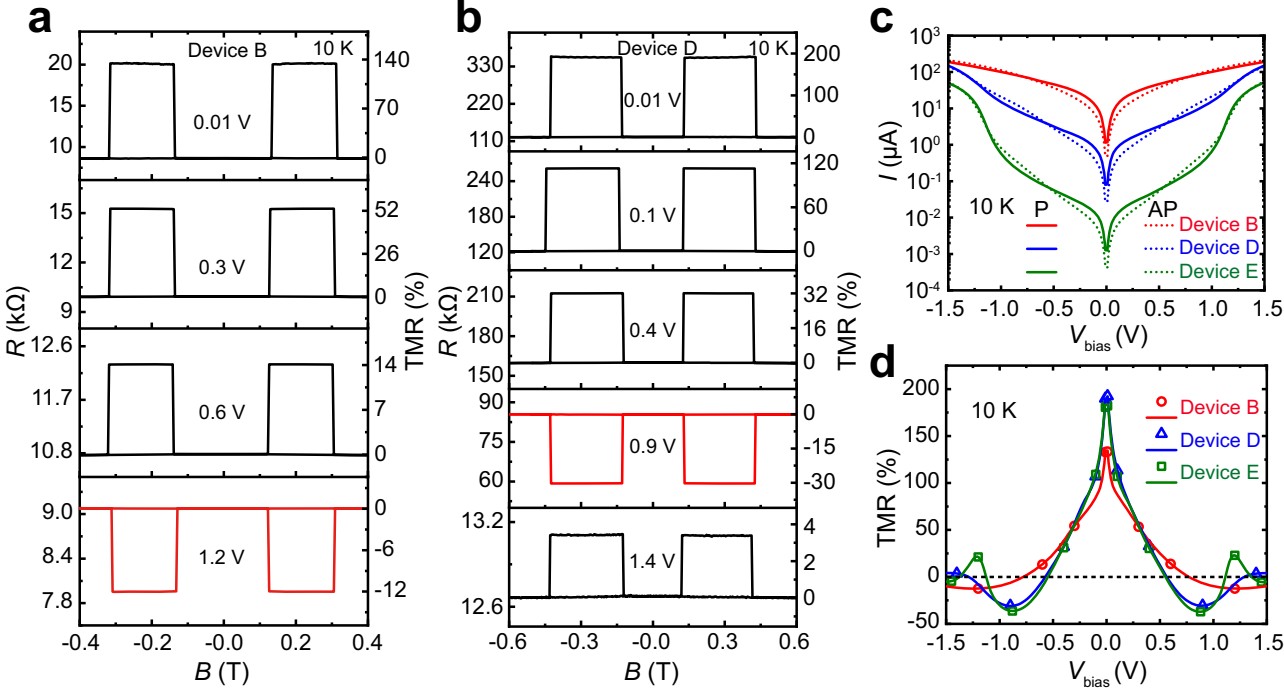

**Fig. 2 | The bias-dependent TMR of the devices. a, b** The $R–B$ curves at various positive bias for devices B and D. **c** $I–V_{bias}$ curves of devices B, D, and E in parallel and antiparallel states, respectively. **d** The corresponding TMR as a function of $V_{bias}$. The hollow symbols are extracted from the $R–B$ curves. The temperature is fixed at 10 K.

shown in a larger current range (inset of Fig. 1b). The nonlinear $J–V_{bias}$ characteristics as well as the obvious parabolic $R–V_{bias}$ curves (Supplementary Fig. 3) reveal the existence of the tunnel barrier between GaSe and FGTs. The band alignment at the FGT/GaSe interface, obtained by analyzing the transport mechanism[42,43], also reveals that the effective tunnel barrier of electrons is up to ~0.9 eV (Supplementary Fig. 4). In some previous studies, the effective tunnel-barrier heights of CrI$_3$/hBN, Fe/MgO and CoFe/MgO interfaces are estimated to be 0.25 eV[44], 0.39 eV[4], and 0.9 eV[45], respectively. Therefore, the tunnel-barrier height of 0.9 eV in the FGT/GaSe interface could ensure the tunneling transport of electrons.

We next examine the TMR. Upon sweeping the out-of-plane **B**, the devices A–F show two distinct parallel ($R_P$) and antiparallel resistance ($R_{AP}$) states (Supplementary Fig. 5). Among them, at bias of 10 mV, the $R_P$ and $R_{AP}$ of device D are 122.25 kΩ and 357.52 kΩ, respectively (Fig. 1c). The corresponding TMR $= (R_{AP} − R_P)/R_P$ is 192.4%. The thickness of GaSe-layer dependence of the measured maximum TMR ratio is shown in Fig. 1d, which first increases from 83.1% (device A) and 133.5% (device B) and 180.2% (device C) to 192.4% (device D) and then decreases to 182.3% (device E) and 121.7% (device F), and finally vanishes (devices G) with increasing the thickness of GaSe layer. The zero-bias resistance-area product ($RA$) of the devices for both the parallel and antiparallel states increases approximately exponentially with increasing the GaSe thickness, suggesting that the transport mechanism is dominated by tunneling (inset of Fig. 1d)[46]. The large TMR and nonlinear $J–V$ curve indicate that the GaSe spacer serves as a good tunnel barrier. The maximum TMR at low bias is found in the device with GaSe of 8.2 nm, and the internal physical mechanism can be explained as follows. On the one hand, the spin-filtering effect of GaSe gets weaker when the thickness is reduced, resulting in the decrease of TMR[2,13]. On the other hand, with further increase of the spacer thickness, the TMR decreases and, eventually, vanishes in device G with 15.6-nm-thick GaSe, where the tunneling through extrinsic defects could become important and eventually exceed the spin-relaxation length of GaSe[47,48]. To confirm the repeatability of this barrier-thickness-dependent TMR behavior, we

we fabricated and measured another group of devices with variable barrier thickness (Supplementary Fig. 6). Similar layer-thickness-dependent TMR was proved, further confirming that the magnitude of TMR is directly related to the semiconductor layer thickness, which can be improved by optimizing the thickness and quality of the barrier layer.

To investigate TMR($V_{bias}$) for devices with different GaSe thickness, we measured the $R–B$ curves. As shown in Fig. 2a, b, the positive TMRs decrease with $V_{bias}$. Negative TMRs of −12.3% and −30.5% for devices B and D are obtained at 1.2 V and 0.9 V, respectively. This salient sign inversion of TMR is found in all the devices A–F. The number of sign reversals of TMR can be tuned, from single to multiple, with increasing the GaSe thickness. To better understand the variation of TMR with bias, as shown in Fig. 2c, we measured the $I–V_{bias}$ curves of the devices in parallel and antiparallel states, respectively. The nonlinear $I–V_{bias}$ curves for devices B, D, and E show very different trends in parallel and antiparallel states, which allow us to derive bias-dependent TMR for these devices (Fig. 2d). The obtained TMR value matches well to that extracted from the $R–B$ curves (Fig. 2a, b and Supplementary Figs. 7 and 8), indicating the influence of the Zeeman effect on TMR is negligible. The symmetric bias-dependent current and TMR suggest the symmetrical FGT/GaSe interfaces in these devices.

Devices A (Supplementary Fig. 9) and B show similar bias-dependent behavior of TMR. Specifically, as shown in Fig. 2c, for device B, the measured current for the parallel state is higher than that for the antiparallel state for $V_{bias} < 0.76$ V, leading to a positive TMR. However, beyond such $V_{bias}$, the measured current for the parallel state is lower than that for the antiparallel state, resulting in a negative TMR (Fig. 2d). With increasing the thickness of the GaSe spacer layer, we observed multiple sign changes of the TMR. As shown in Fig. 2c, the nonlinear $I–V_{bias}$ curves of device D in parallel and antiparallel states show two crossovers, and similar behavior is also observed in device C (Supplementary Fig. 10). Correspondingly, in Fig. 2d, the TMR of device D first decreases monotonically and changes sign of around 0.58 V, and again around 1.27 V as the bias increases. With further increasing the GaSe thickness, as shown in Fig. 2c, the nonlinear $I–V_{bias}$

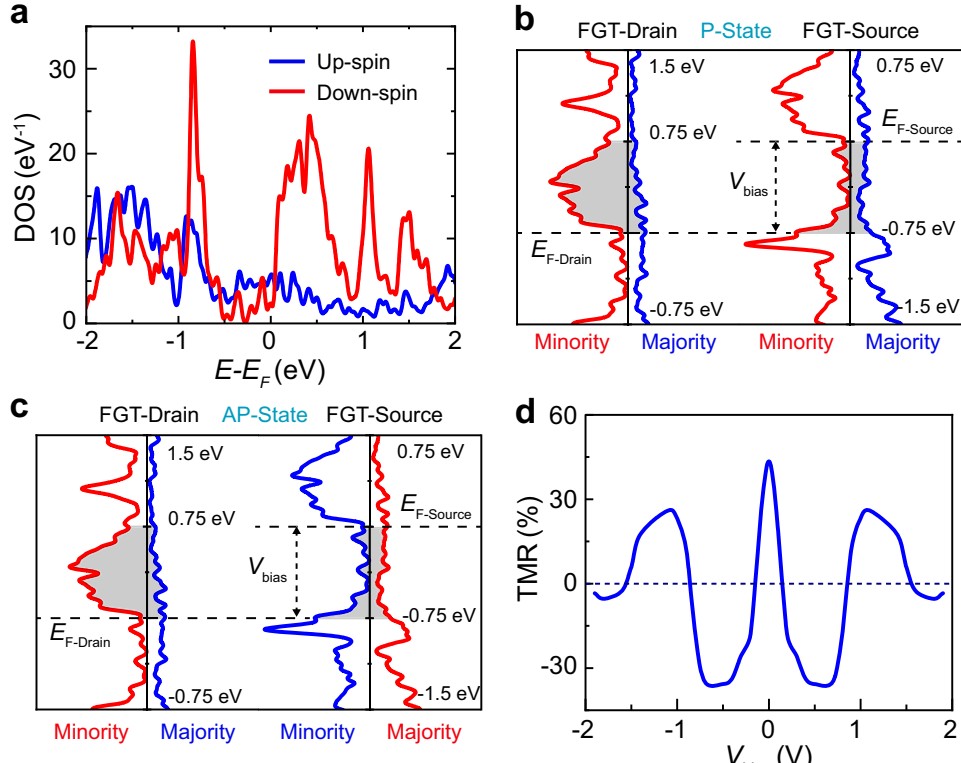

**Fig. 3 | The simulation of spin-resolved DOS of FGT and bias-dependent TMR calculated by the elastic tunneling model. a** The calculated spin-resolved DOS for both the up (blue line) and down (red) spins of the FGT in 3-layer-FGT/6-layer-GaSe/3-layer-FGT heterojunction. **b, c** Schematic diagram of direct band-to-band spin-dependent tunneling under bias window $V_{bias}$ (shaded area) in the parallel and antiparallel states, respectively. **d** The calculated TMR as a function of $V_{bias}$ by using the simple elastic tunneling formula.

curves reveal three crossovers for device E. In Fig. 2d, the TMR of device E first decreases monotonically and changes sign around 0.56 V, and then the TMR decreases in an oscillatory fashion as the bias increases. Similar three-sign changes of TMR behavior is also observed in device F (Supplementary Fig. 11).

To understand the bias-dependent magnetotransport, the spin-resolved density of states (DOS) of FGT in the 3-layer-FGT/6-layer-GaSe/3-layer-FGT heterojunction were obtained using the first-principles calculations (see Supplementary Fig. 12). The calculated spin-resolved DOS of FGT electrode is shown in Fig. 3a. Assuming that tunneling is elastic and spin-conserving, with the corresponding tunneling probability independent of the initial and final states as well as the tunnel-barrier height for electrons[49], the spin-dependent tunneling current at zero temperature can be expressed as[1,2,50]

$$I_{\sigma} \propto \int_{\mu_D}^{\mu_S} \rho_D^{\sigma}(E - \mu_D) \rho_S^{\sigma}(E - \mu_S) dE, \tag{1}$$

where the $\mu_{D(S)}$ and $\rho_{D(S)}^{\sigma}$ are the chemical potentials and spin-resolved DOS for the drain (or source) FGT, respectively ($\sigma$ is the spin index), and $\mu_D - \mu_S = eV_{bias}$. The tunneling currents in parallel (Fig. 3b) and antiparallel (Fig. 3c) configurations can thus be expressed as

$$I_P \propto \int_{\mu_D}^{\mu_S} \left( \rho_D^{\uparrow}(E - \mu_D) \rho_S^{\uparrow}(E - \mu_S) + \rho_D^{\downarrow}(E - \mu_D) \rho_S^{\downarrow}(E - \mu_S) \right) dE, \tag{2}$$

and

$$I_{AP} \propto \int_{\mu_D}^{\mu_S} \left( \rho_D^{\uparrow}(E - \mu_D) \rho_S^{\downarrow}(E - \mu_S) + \rho_D^{\downarrow}(E - \mu_D) \rho_S^{\uparrow}(E - \mu_S) \right) dE. \tag{3}$$

The resulting bias-dependent TMR is shown in Fig. 3d. With increasing bias, the calculated TMR rapidly drops, changes sign, and then oscillates in qualitative agreement with the measurements for devices E and F. Multiple sign changes of TMR with increasing bias have also been predicted theoretically for Fe/MoS$_2$/Fe MTJ devices[13]. In addition, the multiple sign changes of TMR have also been observed in FGT/hBN/FGT, FGT/WSe$_2$/FGT, and FGT/WS$_2$/FGT MTJs[27]. In our FGT/GaSe/FGT MTJs, within the range of 1.5 V bias, as the thickness of a GaSe spacer decreases, the number of TMR reversals decreases to two in devices C and D and to only one in devices A and B. A possible physical mechanism of the thickness-dependent TMR reversals is explained as follows.

The above calculation of the spin-dependent tunneling current assumed that the transmission probability is the same for all initial and final states. However, if tunneling is at least partially coherent, the transmission probability should be larger for states with transverse momenta close to those where the decay rate of the evanescent states in GaSe is the smallest. Because the bandgap in GaSe is indirect, this may occur away from the Gamma point[51]. The efficiency of this transverse-momentum filtering increases with increasing thickness of the tunnel barrier[50].

Because the interlayer dispersion of the vdW FGT states is weak, they are at a given transverse momentum essentially quantized. An energy isosurface of a lead thus consists of one or more 1D Fermi contours. Coherent tunneling is only possible at the intersections of the isosurfaces of the two leads corresponding to the same electrochemical potential. At large barrier thickness, a strong enhancement of the tunneling current should occur when such crossing points fall close to the points of the lowest decay rate in GaSe. If such matching condition is satisfied at the given bias for initial and final states of the

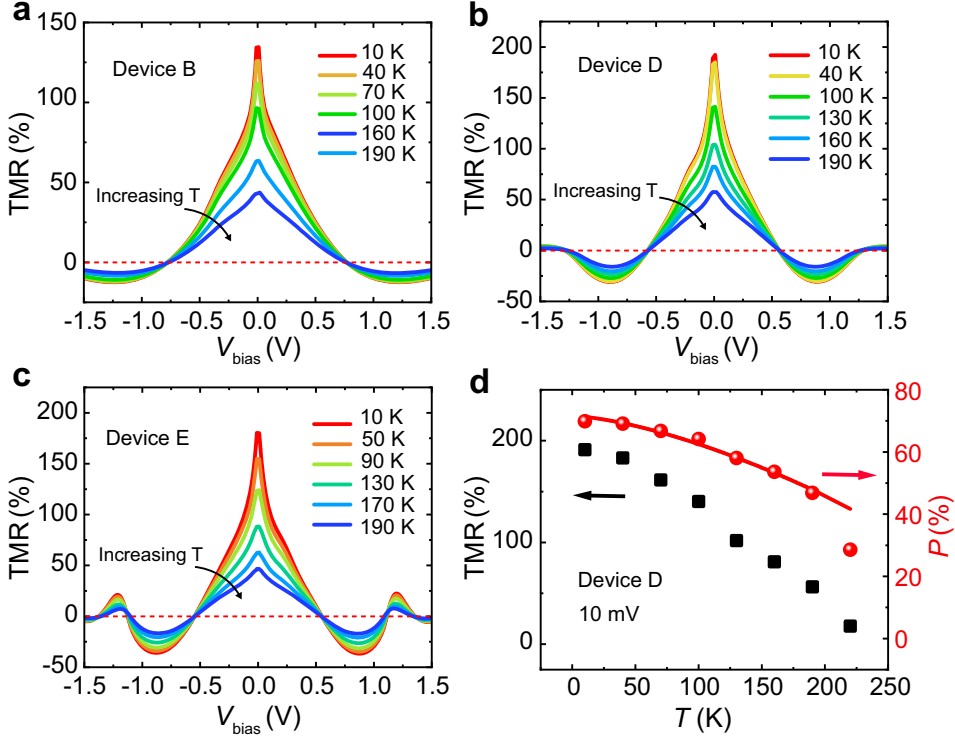

**Fig. 4 | The temperature-dependent TMR. a–c** TMR ratios of devices B, D, and E measured at temperatures from 10 to 190 K, respectively. **d** The TMR and spin polarization of device D as a function of temperature at bias of 10 mV. The red line shows the fitting data by the Bloch's law[1].

same or opposite spin, positive or negative TMR is expected, respectively (Supplementary Fig. 13). The relative importance of such matching should increase at a larger barrier thickness, as long as the coherent tunneling persists. Thus, coherent tunneling may explain why additional sign changes of TMR as a function of bias are observed at larger GaSe thicknesses.

On the other hand, the barriers thickness-dependent TMR reversals can be related to the tunnel junction resistance values. With semiconducting tunnel barriers as thick as the GaSe film in device E, the elastic tunneling model with FGT electronic structures can be applicable since all the voltage drops of the device should be happening at the tunnel junctions. With thinner semiconducting films and lower tunnel junction resistances, however, the bias voltage effectively applied to the tunnel junctions should be smaller than the voltage driven by an external voltage source: this phenomenon is often dubbed as a voltage-divider effect. To prove this, we extracted the bias voltages where the TMR-sign reversals firstly occur for different devices (Supplementary Fig. 14), the bias voltages increases as the tunnel junction resistance decreases, indicating the voltage-divider effect could also be the reason for the barriers thickness-dependent TMR reversals.

We further investigate the temperature-dependent TMR effect in our devices. The $I-V_{bias}$ curves of devices B, D, and E in parallel and antiparallel states were measured at temperatures from 10 K to 190 K. The corresponding TMR is plotted in Fig. 4a–c for device B, D, and E, respectively. The TMR-$V_{bias}$ curves in Fig. 4a–c all pass through the same zero at all temperatures, verifying that the bias-dependent TMR in the devices is dominated by tunneling[42]. The extracted TMR at different temperatures at 10 mV for device D (Fig. 4d) and device B and E (Supplementary Fig. 15) show a decreasing trend, which can be attributed to the decrease of spin polarization with temperature. When $V_{bias}$ is extremely small, approximately only the electrons at the $E_F$ take part in tunneling transport. For simplicity, assume the source and drain FGT electrodes have almost the same

spin polarization. Then TMR can be defined as TMR $= 2P^2/(1 - P^2)$, where $P$ denote the spin polarization at the $E_F$ for the drain and source FGT electrode[1,2]. As shown in Fig. 4d, the $P$ decreases with temperature, and the maximum $P$ at 10 K is up to 70%, which is larger than that obtained in other 2D semiconductor-based MTJs, such as 45% in Fe₃GeTe₂/InSe and 60% in Fe₃GeTe₂/WSe₂ interfaces[27,28], but lower than that of Ni(111)/Gr interface[52]. The estimated temperature-dependence of the spin polarization can be fitted well by Bloch's law, given by $P = P_0(1 - \alpha T^{3/2})$, where $P_0$ is the spin polarization at 0 K, $\alpha$ is a materials-dependent constant[53]. The fitting value of $\alpha$ is $1.26 - 1.40 \times 10^{-4} \, K^{-3/2}$, which is comparable to the previous reports[26,40].

Our presented results, which demonstrate large and tunable TMR, substantiate an ambitious vision where all-vdW MTJs could replace various charge-based memory applications[5], targeted to reach TMR ~200% for commercial viability. Implementing such vdW MTJs is expected to rely on insulating 2D tunnel barrier[5], just as it was shown with hBN barrier in an all-vdW MTJ with the low-temperature TMR ~160%[26] and also supported by a very recent report of TMR ~300%[27]. However, since we observe the desired large TMR values even with a semiconductor spacer, the prospect for all-vdW spintronics becomes considerably broader than just memory applications and the resulting large spin polarization and spin–orbit coupling opens opportunities beyond magnetoresistive effects. For example, the measured sign reversal of the TMR with applied bias is consistent with the reversal of the carrier spin polarization and could enable desirable polarization modulation[36]. Our findings could integrate semiconductor-based optoelectronics, microelectronics, and spintronics together, and be also relevant to emerging cryogenic applications where proximity-modified semiconductors and MTJs provide a platform for fault-tolerant quantum computing[54]. Given the continued advances in understanding of vdW semiconductor/ferromagnet junctions, including a surprising optical manifestation of the valley-dependent magnetic proximity effect[55,56], we expect further opportunities for valley-

dependent transport properties by using semiconducting tunnel barriers in MTJs.

## Methods

**Fabrication of the Fe₃GeTe₂/GaSe/Fe₃GeTe₂ MTJ devices.** The high-quality vdW bulk single-crystal FGT and hBN were purchased from HQ Graphene, while GaSe was purchased from 2D semiconductors, respectively. Firstly, a FGT flake was exfoliated onto polydimethylsiloxane (PDMS) stamps by adhesive tape. The stamps were adhered to a glass slide. Under optical microscope, the FGT flake with appropriate thickness and shape was chosen to transfer onto a 300-nm-thick $SiO_2$/Si substrate by using a position-controllable dry transfer method[40]. Then, using the same method, a GaSe flake was transferred onto the FGT flake, followed by another thicker FGT flake to fabricate a 2D heterojunction. To prevent the FGT from oxidation, a 20 nm-thick hBN layer was used to cap the whole heterostructure stack. Finally, the device was annealed at 120 °C for 10 min to remove the bubbles between the layers and ensure close contact between the layers. Notably, the whole transfer processes were performed in a nitrogen-filled glovebox with a concentration of less than 1 ppm of oxygen and water to ensure a clean interface. The source and drain electrode regions were pre-patterned by standard photolithography, and Cr/Au (10/40 nm) layers were deposited using an ultrahigh vacuum magnetron sputtering system, followed by a lift-off process. The thicknesses of both GaSe and FGT flakes were measured by using AFM (Bruker Multimode 8).

**Measurements of MR effect.** Electrical properties were measured using a semiconductor characterization system (Agilent Technology B1500A). All measurements were carried out in a Model CRX-VF Cryogenic Probe Station with a ± 2.5 T out-of-plane vertical magnetic field.

**Ab Initio Calculations.** Our first-principles calculations are performed by the density functional theory (DFT) using the Vienna ab initio Simulation Package (VASP) code. The details are shown in the Supplementary Note 7.

## Data availability

The data that support the findings of this study are available within the article and the Supplementary Information or available from the corresponding author upon reasonable request. All data generated in this study are provided in the Supplementary Information/Source Data file. Source data are provided with this paper.

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

## Acknowledgements

This work was financially supported by the National Key Research and Development Program of China (Grants No. 2022YFA1405100, K.W.), the Beijing Natural Science Foundation Key Program (Grant No. Z190007, K.W.), the National Natural Science Foundation of China (Grants No. 12241405, K.W. and No. 12174405, H.Y.), the Strategic Priority Research Program of Chinese Academy of Sciences (Grants No. XDB44000000 and No. XDB28000000, K.W.), and the National Science Foundation under Grants No. ECCS-2130845 (I.Ž.) and No. DMR-1916275 (K.B.).

## Author contributions

K.W. conceived the work. W.Z. fabricated the devices, W.Z., F.Y., Y.D., H.L., and Z.W. performed the experiments. W.Z., K.B., and K.W. analyzed the data. Y.Z., T.Z., X.Z., Q.C., H.Y., I.Ž., K.B., and K.W. carried out the modeling. W.Z., H.Y., L.Z., I.Ž., K.B., and K.W. wrote the manuscript. All authors discussed the results and commented on the manuscript.

## Competing interests

The authors declare no competing interests.
