## [Peer Review File · Nature Communications]

Reviewers' Comments:

Reviewer #1:

Remarks to the Author:

Comments to the authors:

The authors have fabricated all van der Waals (vdW) assembled magnetic tunnel junctions (MTJs) with two-dimensional metallic ferromagnet Fe₃GeTe₂ (FGT) and semiconducting GaSe as a tunnel insulator. They report that tunneling magnetoresistance (TMR) ratios from the FGT–GaSe–FGT MTJ devices become significantly enhanced up to ~ 192 % for the device with a 9-layer GaSe spacer, and the TMRs can be directly related to the layer number of the semiconducting GaSe films. They have stressed that sample-bias (V_{bias}) dependent spin-valve operations, particularly the number of the TMR-sign changes, can also be associated with the GaSe thickness.

The authors claim that the GaSe-thickness-dependent spin-valve operations and the implementation of semiconducting GaSe as a tunnel insulator make the current manuscript unique and worthy for considering publication in Nature Communications when compared with other previous reports on vdW MTJs, particularly the very recent study by Min et al. (Ref 34): although they claim that their FGT–WSe₂–FGT MTJ operations are not related to tunnel-insulator WSe₂ thicknesses. Experimental realizations of vdW MTJs and spintronic device applications with various materials are always welcome in spintronic communities. However, experimental verifications and theoretical supports made in the current manuscript for their primary GaSe-thickness-dependent spin-valve operations are not at a satisfactory level, along with serious doubts in data interpretations, as listed below. Thus, I do not support the publication of the current manuscript in Nature Communications until the following questions/issues are fully resolved.

1. Of various two-dimensional semiconductor vdW materials, the authors have exclusively utilized GaSe as a tunnel barrier for their MTJs. Moreover, the central claim made in the current manuscript is deeply associated with the layer-number-dependent electronic properties of the GaSe. In the current manuscript, however, the authors do not elaborate on why they chose GaSe as the tunnel insulator, along with all the necessary details of the semiconducting GaSe films, such as basic electronic structures, band gaps, semiconducting types, and others.

2. The authors claim that semiconducting barriers in MTJs are superior to other insulating spacers for achieving high TMR ratios, which requires detailed explanations with proper references. I see some advantages of implementing semiconducting layers as spacers; for example, as the authors have stated in the abstract, gate-tunable spin-device operations are possible. As far as TMR values are concerned, however, no immediate advantages of small energy-gap materials (semiconducting materials) over wide energy-gap materials (insulating materials) are anticipated.

3. The GaSe layer-number-dependent transport characteristics (Fig. 1b) and the consequent TMR variations (Fig. 1d) require further explanations. The authors have fabricated six (seven in total) measurable FGT–GaSe–FGT MTJs with GaSe layer numbers varying from six to eleven, whose corresponding thicknesses vary from 5.5 nm to 10.0 nm (they assume the thickness of single-layer GaSe films ≈ 0.92 nm). With these thickness variations, however, the transport characteristics presented in Fig. 1b seem widely dispersed. For example, charge flows through devices A (six layers) and B (seven layers) are like transports through either shorted or shallow tunnel-gap junctions. Devices C (eight layers) and D (nine layers) reveal reasonable tunnel-behavior characteristics, although discernible Fowler-Nordheim features are nonexistent. With one or two more GaSe layers, however, J–V relations at devices E (ten layers) and F (eleven layers) are similar to classic transport characteristics curves for large tunnel-gap junctions, and the authors also claimed that the devices have a maximum tunnel barrier up to ~ 0.83 eV. Are there any reasons their MTJs are highly sensitive to the layer number of GaSe, simply changing from six to eleven? Is the energy gap of GaSe layers, thus the tunnel-barrier height susceptible to the change of GaSe layer number from six to eleven? As far as I know, no semiconducting materials reveal the electronic structure variations as conspicuous as the GaSe films in the current study when the film thickness changes from 5 nm to 10 nm. Therefore, I request that they need to prepare another set of FGT–GaSe–FGT devices with similar GaSe thicknesses to confirm their

primary findings in the current manuscript.

4. The authors claim that additional sign changes of TMRs can be related to the spin-polarized electronic structures of FGTs (Fig. 3a and 3b), and the multiple TMR sign changes are reproduced with spin-dependent tunneling models with considerations of chemical potentials and spin-dependent density of states (DOS) of FGTs. To be physically acceptable for applying the tunneling model, however, tunnel-barrier heights for the MTJs should be much higher than the chemical potentials of FGT-source and FGT-drain electrodes (Fig. 3b), allowing the electron tunnelings to be governed by the direct tunneling mechanism, just like the FGT-hBN-FGT device in the Ref 31. However, FGT-GaSe-FGT devices have much lower tunnel barrier heights (< 0.83 eV), which raises serious doubts about applying the tunnel model to explain the GaSe-layer-number-dependent TMR features, not to mention the ambiguities of explaining how the number of TMR-sign reversals are linked to the thickness of GaSe spacer.

Reviewer #2:

Remarks to the Author:

The manuscript by Zhu et al. entitled « Large and tunable magnetoresistance in van der Waals Ferromagnet/Semiconductor junctions» addresses the topic of magnetic tunnel junctions with 2D-SC and FGT electrodes. The article shows very interesting results of MR for a system with two FGT electrodes and a GaSe tunnel barrier.

I overall find the results very interesting and encouraging. The observation of high MR at low temperature in such structure is inspiring for the community, opening new opportunities. With the following few points being clarified (notably the differences with ref Nat. Mater. 21,1144 (2022) already cited in the manuscript) this article will be a great contribution to the field and I would gladly recommend its publication in Nature Communications.

Here are my comments/questions to the authors:

Line 28-41 Overall the abstract and its claims could win from clarification. In the abstract, the authors say that the "Integration of two-dimensional ferromagnets in semiconductor-based vdW junctions offers gate-tunability, bias dependence, magnetic proximity effects, and spin-dependent optical-selection rules." This sentence could be clarified as it is not clear what property is expected to come from the ferromagnet and what is expected to come from the semiconductor. Typically, the semiconductors can't be gated within the junction and probably optical selection rules apply to the semiconductor... Also, the author should split the sentence "We demonstrate that not just the magnitude, but also the TMR sign is tuned by the applied bias or the semiconductor thickness, enabling modulation of highly spin-polarized carriers in vdW semiconductors." as the TMR sign is not tuned by the thickness but by the applied bias while going to higher voltages. The authors should also specify that samples are based on flakes.

Line 49 In the sentence "MTJs with both conventional or vdW ferromagnets typically include an insulating spacer layer, which makes it easier to achieve high TMR compared to a semiconducting barrier." Could the authors comment on why insulators should intrinsically give a lower MR than SC ? As it seems it is not the presence of the insulator per se that gives high MR but more the specific filtering mechanism linked to certain ones such as MgO.

Line 55-58 Here if graphene is included in the discussion the work on 2DvdW could also include a more broaden literature (such as Phys. Rev. B 90, 085429 (2014), Sci. Rep.6, 21168 (2016), 2D Mater. 4 1014001 (2017), ACS Nano, 12, 4712 (2018), 2D Mater. 7, 015026 (2020), Nat Commun 11, 3657 (2020), Nat. Commun. 11 5670 (2020), Commun. Phys. 4, 124 (2021)...)

Line 64, reference should be made to Nat. Mater. 21,1144 (2022) for which 110% was found with WSe2.

Line 85 Fig 1a. A picture of the real structure is missing. Could the authors include in the core text

an actual image of the device where the different layers can be clearly distinguished alike Figure S1 (only figure S1 is too low resolution).

Line 72 While the text gives a certain amount of details on the GaSe layers, information on the FGT layers are missing: like the thicknesses of both FGT layers in the devices.

Line 79 Fig 1b. The J-V curve for devices C and D which are the ones showing the highest MR do not really look like conventional non-linear tunneling curves (parabolic G-V) with some kind of kink at low voltage. Could the authors comment on that ?

Line 114. The authors suggest that the MR can be improved by optimizing the thickness but this seems counterintuitive when looking at Fig1d. There, 200% seems to be the maximum. Also, Line 110 what exact mechanism do the authors involve to explain the reduction at low barrier thickness ?

Line 153 Fig3b appears puzzling to me and could win to be clarified. Why are the authors considering that P state carriers correspond to the lowest energy (E_f -drain) and AP state to the highest (E_f -source)? According to the picture given by the authors, carriers are tunneling from the right electrode to the left. More importantly the ones with lower tunnel barrier height, hence higher energy(at E_f -source), are the one that contribute the most to the total current. I would thus mainly expect carriers corresponding to the green arrows to dominate the current. What are the blue ones corresponding to ? Importantly, Fig2b and Fig3c and Fig4 look a lot like figures from Nat. Mater. 21,1144 (2022). Reference/comparison to this work should be mentioned in the discussion.

Line 180 As core to the paper, the model could benefit from further discussion and explanation of the physical mechanism at play.

Also, it was suggested in Nat. Mater. 21,1144 (2022) that the reversal of MR as a function of bias was occurring for h-BN, WSe2 and WS2. The authors should compare their result with this work.

Line 186 An experimental demonstration involving indirect band gap with states away from the gamma point in 2D SC was previously discussed (ACS Nano 13, 14468 (2019))

Line 190 Have the authors been calculating such matching ? If yes, could they provide some feedback.

Line 217 and 226 the authors demonstrate a spin polarization of 70%, it is worth to compare it with other results on other 2D SC based MTJs with different barriers such as Nat. Mater. 21,1144 (2022) and even other 2Ds such as the -98% observed for graphene ACS Nano 16, 14007 (2022).

Fig S2 raises some questions. Why can one observe a jump in FigS2c for qPhy, whereas it is not seen at all in Fig1c, in either the TMR or R.A vs thickness plots.

Reviewer #1 (Remarks to the Author):

Comments to the authors:

The authors have fabricated all van der Waals (vdW) assembled magnetic tunnel junctions (MTJs) with two-dimensional metallic ferromagnet Fe_3GeTe_2 (FGT) and semiconducting GaSe as a tunnel insulator. They report that tunneling magnetoresistance (TMR) ratios from the FGT–GaSe–FGT MTJ devices become significantly enhanced up to $\sim 192\%$ for the device with a 9-layer GaSe spacer, and the TMRs can be directly related to the layer number of the semiconducting GaSe films. They have stressed that sample-bias (V_{bias}) dependent spin-valve operations, particularly the number of the TMR-sign changes, can also be associated with the GaSe thickness.

The authors claim that the GaSe-thickness-dependent spin-valve operations and the implementation of semiconducting GaSe as a tunnel insulator make the current manuscript unique and worthy for considering publication in Nature Communications when compared with other previous reports on vdW MTJs, particularly the very recent study by Min et al. (Ref 34): although they claim that their FGT– WSe_2 –FGT MTJ operations are not related to tunnel-insulator WSe_2 thicknesses. Experimental realizations of vdW MTJs and spintronic device applications with various materials are always welcome in spintronic communities. However, experimental verifications and theoretical supports made in the current manuscript for their primary GaSe-thickness-dependent spin-valve operations are not at a satisfactory level, along with serious doubts in data interpretations, as listed below. Thus, I do not support the publication of the current manuscript in Nature Communications until the following questions/issues are fully resolved.

1. Of various two-dimensional semiconductor vdW materials, the authors have exclusively utilized GaSe as a tunnel barrier for their MTJs. Moreover, the central claim made in the current manuscript is deeply associated with the layer-number-dependent electronic properties of the GaSe. In the current manuscript, however, the authors do not elaborate on why they chose GaSe as the tunnel insulator, along with all the necessary details of the semiconducting GaSe films, such as basic electronic structures, band gaps, semiconducting types, and others.

Response: We thank the Referee for this insightful suggestion. First, we will explain in detail why GaSe is chosen as the tunnel barrier. Then we characterize the basic structure, bandgap, and the conductivity of GaSe in detail.

In our previous work, we have studied the magnetoresistance properties of a series of Fe_3GeTe_2 (FGT)-based spin-valve devices with different spacer layers, such as FGT/MoS₂/FGT, FGT/WS₂/FGT, FGT/WSe₂/FGT, and FGT/InSe/FGT junctions [H. Lin et al. *ACS Appl. Mater. Inter.* 12, 43921 (2020); C. Hu et al. *Chin. Phys. B* 30, 97505 (2021); Y. Zheng et al. *npj 2D Mater. Appl.* 6, 62 (2022); W. Zhu et al. *Adv. Mater.* 33, 2104658 (2021)]. We find the magnetoresistance is sensitive to the transport mechanism in which the tunneling devices show a much larger magnetoresistance than that of the Ohmic transport devices. 2D gallium selenide (GaSe) crystal is a typical layered-metal monochalcogenide with an indirect bandgap energy of ~ 2 eV in the bulk, which has been used in the field of ultraviolet photoelectric detection in our previous works [Y. F. Cao. *Sci. Rep.* 5, 8130 (2015); F. G. Yan et al. *Nanotechnology* 28, 27LT01 (2017)]. GaSe has also been proved to be a good tunnel barrier material [S. Kurtin et al. *Phys. Rev. Lett.* 25, 756 (1970)]. Therefore, we expect to have a larger magnetoresistance in $\text{Fe}_3\text{GeTe}_2/\text{GaSe}/\text{Fe}_3\text{GeTe}_2$ devices. We add the related information to the introduction of the main manuscript.

The basic properties of GaSe have been investigated in our pervious works [Y. F. Cao. *Sci. Rep.* 5, 8130 (2015); F. G. Yan et al. *Nanotechnology* 28, 27LT01 (2017)]. Gallium selenide (GaSe) single crystals are a typical 2D layered metal monochalcogenide with an indirect bandgap energy of ~ 2 eV in the bulk [Y. F. Cao et al. *Sci. Rep.* 5, 8130 (2015)]. To characterize the crystal quality of GaSe, we measured the Raman and PL spectrum of the bulk GaSe. As shown in **Figure R1a**, the Raman spectrum of the bulk GaSe shows peaks at ~ 135 cm^{-1} , 214 cm^{-1} , 245 cm^{-1} , and 308 cm^{-1} , corresponding to the A_{1g}^2 , E_{2g}^1 , E_{1g}^2 , and A_{1g}^2 vibration mode of GaSe, respectively. The PL measurements indicates the bandgap energy of the bulk GaSe is ~ 2 eV (**Figure R1b**). These notations are consistent with earlier works on bulk ϵ -GaSe [X. Li et al. *Sci. Rep.* 4, 5497 (2017); Maciej R. Molas et al. *Faraday Discuss.* 227, 163 (2021)]. We add the related information in the Supplementary Information (Note 1 and Figure S1).

Figure R1. (a) The Raman spectrum and (b) Photoluminescence (PL) spectrum of a bulk GaSe flake (measured with 532 nm laser excitation at room temperature). The spectrum shows the band edge emission of GaSe, which is centered at about 2 eV.

2. The authors claim that semiconducting barriers in MTJs are superior to other insulating spacers for achieving high TMR ratios, which requires detailed explanations with proper references. I see some advantages of implementing semiconducting layers as spacers; for example, as the authors have stated in the abstract, gate-tunable spin-device operations are possible. As far as TMR values are concerned, however, no immediate advantages of small energy-gap materials (semiconducting materials) over wide energy-gap materials (insulating materials) are anticipated.

Response: We thank the Referee for this valuable comment and the opportunity to explain the relevance of semiconductor-based tunnel barriers. Compared with insulator, the advantage of semiconductor tunnel barrier is that its Fermi level (E_F) can be adjusted by doping to make the Fermi level close to the valence band or closer to the conduction band [K. Dolui *et al. Phys. Rev. B* 90, 041401 (2014)]. On the one hand, if E_F lies close to the conduction band minimum, a K-point in Brillouin zone (BZ) dominated transmission is obtained from evanescent states originating from the conduction band. Therefore, transport in thicker junctions at E_F is conduction band dominated. On the other hand, if E_F lies close to the valence band maximum, a Γ -point in BZ dominated transmission is obtained, which leads to the transport being controlled by the valence band. This controllable transport plays an important role in enhancing spin-filtering effect. We add the related information to the introduction of the main manuscript.

3. The GaSe layer-number-dependent transport characteristics (Fig. 1b) and the consequent

TMR variations (Fig. 1d) require further explanations. The authors have fabricated six (seven in total) measurable FGT–GaSe–FGT MTJs with GaSe layer numbers varying from six to eleven, whose corresponding thicknesses vary from 5.5 nm to 10.0 nm (they assume the thickness of single-layer GaSe films ≈ 0.92 nm). With these thickness variations, however, the transport characteristics presented in Fig. 1b seem widely dispersed. For example, charge flows through devices A (six layers) and B (seven layers) are like transports through either shorted or shallow tunnel-gap junctions. Devices C (eight layers) and D (nine layers) reveal reasonable tunnel-behavior characteristics, although discernible Fowler-Nordheim features are nonexistent. With one or two more GaSe layers, however, J–V relations at devices E (ten layers) and F (eleven layers) are similar to classic transport characteristics curves for large tunnel-gap junctions, and the authors also claimed that the devices have a maximum tunnel barrier up to ~ 0.83 eV. Are there any reasons their MTJs are highly sensitive to the layer number of GaSe, simply changing from six to eleven? Is the energy gap of GaSe layers, thus the tunnel-barrier height susceptible to the change of GaSe layer number from six to eleven? As far as I know, no semiconducting materials reveal the electronic structure variations as conspicuous as the GaSe films in the current study when the film thickness changes from 5 nm to 10 nm. Therefore, I request that they need to prepare another set of FGT–GaSe–FGT devices with similar GaSe thicknesses to confirm their primary findings in the current manuscript.

Response: We thank the Referee for these valuable comments. The dispersed J - V curves can be attributed to the different conduction mechanisms in devices with different GaSe thickness. We carefully re-analyze the J - V curves for the devices with different GaSe spacer and find the current tunneling mechanism is dependent on the GaSe thickness.

The specific steps of this analysis are as follows.

When the applied bias voltage is far less than the effective tunnel-barrier height ($V_{\text{bias}} \ll \phi_B$), the conduction mechanism is only direct tunneling, and current density-voltage dependence of direct tunneling is described by the Simmons' formula [*J. G. Simmons, J. Appl. Phys. 34, 1793 (1963)*], that is,

$$J = \frac{3q^2(2m^*q\phi_B)^{1/2}V_{\text{bias}}}{2h^2d} \exp\left[-\frac{4\pi d(2m^*q\phi_B)^{1/2}}{h}\right],$$

where q is the elementary charge, d is the barrier width, m^* is the effective electron mass ($\sim 0.1 m_0$ in GaSe [*R. Le Toullec, et al. Phys. Rev. B 22, 6162 (1980)*]), ϕ_B is the effective tunnel-barrier height and h is the Planck's constant. When the applied bias voltage exceeds the effective tunnel-barrier height ($V_{\text{bias}} > \phi_B$), the tunneling mechanism is described by F-N tunneling [*M. Yildirim et al. J. Magn. Magn. Mater. 379, 280 (2015)*], and the F-N tunneling current density can be expressed by the formula of

$$J = \frac{q^3 V_{bias}^2}{8\pi h q \phi_B d^2} \exp \left[\frac{-8\pi d (2qm^*)^{1/2}}{3h V_{bias}} \phi_B^{3/2} \right].$$

To obtain the effective tunnel-barrier height information under different tunneling mechanisms, the above two formulas are linearized with the logarithmic scale, they can be written as:

$$\ln \frac{J}{V_{bias}^2} = \ln \left(\frac{3q^2 (2m^* q \phi_B)^{1/2}}{2h^2 d} \right) + \ln \left(\frac{1}{V_{bias}} \right) - \frac{4\pi d (2m^* q \phi_B)^{1/2}}{h} \text{ for direct tunneling.}$$

$$\ln \frac{J}{V_{bias}^2} = \ln \frac{q^3}{8\pi h q \phi_B t^2} - \frac{8\pi d (2qm^* \phi_B^3)^{1/2}}{3h} \left(\frac{1}{V_{bias}} \right) \text{ for F-N tunneling.}$$

From the J - V_{bias} data measured at 10 K (Figure 1b), we extract $\ln(J/V_{bias}^2)$ versus $1/V_{bias}$ curves for the different devices, as shown in the **Figures R2a-f** below, in which the red dotted line is fitted by the direct tunneling formula. Both devices A and B are dominated by direct tunneling in the bias range of 0-1.5 V. The devices C-F are dominated by direct tunneling under small bias voltage, and gradually show F-N tunneling with the increase of bias voltage. Under small bias voltage, the effective tunnel-barrier heights of the devices are obtained by fitting the Simmons' formula, which decreases slowly with the increase of the barrier thickness, and the effective tunnel-barrier height is ~ 0.9 eV (**Figure R2h**).

With the increase of the barrier thickness, the F-N tunneling become more and more obvious. For F-N tunneling, a plot of $\ln(J/V_{bias}^2)$ versus $1/V_{bias}$ should be linear. In addition, the slope of the F-N tunneling plots can be expressed as a function of the effective mass and the effective tunnel-barrier height: slope = $-6.83 \times 10^9 \times d \sqrt{\left(\frac{m^*}{m_0}\right) \phi_B^3}$. As shown in **Figure R2g**, the F-N tunneling plots of $\ln(J/V_{bias}^2)$ vs $1/V_{bias}$ with narrow abscissa value for the different devices, where the red dotted line is the curve fitted by the F-N tunneling formula. Therefore, the effective tunnel-barrier height for elections of devices C, D, E, F and G can be estimated as 0.42, 0.45, 0.90, 0.88 and 0.86 eV, respectively. The effective tunnel-barrier height is also ~ 0.9 eV except for devices C and D, indicating that the transport mechanism is related to the barrier thickness [*F. C. Chiu, Adv. Mater. Sci. Eng. 578168 (2014)*]. In devices C and D, the direct tunneling current is large and cannot be ignored due to their relatively small barrier thickness. Under large bias voltage, device C and device D are jointly affected by direct tunneling and F-N tunneling. Thus, using the F-N tunneling formula to fit the effective tunnel-barrier heights of device C and device D will lead to deviation. The effective tunnel-barrier height for elections of devices E, F, and G, extracted from the F-N tunneling plots, are shown in the **Figure R2h**. Since the bandgap of bulk GaSe is ~ 2 eV, the energy band diagram of the device under direct and F-N tunneling mechanisms are shown in the inset of **Figure R2h**. We add the related information in the Supplementary Information (Figure S4).

Figure R2. (a-f) The $\ln(J/V_{\text{bias}}^2)$ vs $1/V_{\text{bias}}$ curves for the different devices (blue line), and the red dotted line is fitted by the direct tunneling formula. (g) The F-N tunneling plots of $\ln(J/V_{\text{bias}}^2)$ vs $1/V_{\text{bias}}$ with narrow abscissa value for the different devices, where the red dotted line is the curve fitted by the F-N tunneling formula. (h) The effective tunnel-barrier height is extracted by fitting the direct tunneling formula and F-N tunneling formula respectively. The illustration shows the schematic diagram of direct tunneling under small bias voltage and F-N tunneling mechanism under large-bias voltage. The temperature is fixed at $T = 10$ K.

Thus, by analyzing the J - V curves of the different devices, we can draw the following conclusions. The current of devices A and B is dominated by direct tunneling. The current of devices E and F under larger bias voltage is dominated by F-N tunneling. We agree with the Reviewer that devices C and D have no discernible Fowler-Nordheim features, and their current

could be affected by both direct tunneling and F-N tunneling. We agree with the Reviewer's opinion that the electronic band structure of GaSe films does not change significantly with the thickness from 5 nm to 10 nm. Thus, the effective tunnel-barrier height of the devices is ~ 0.9 eV, which remains unchanged when the thickness of GaSe changes from 5.5 nm to 15.6 nm.

As shown in Figure 1d, a maximum value of TMR has been achieved at a certain barrier thickness of ~ 8 nm. This should be due to the enhancement of spin-filtering effect as the thickness of the barrier layer increases from 5 nm to 8 nm, while with the further increase of the barrier layer thickness, the effect of impurity scattering in the barrier on the tunneling electrons will be enhanced, resulting in the rapid reduction of TMR. In order to confirm the thickness dependent behavior that TMR has a maximum value with varying the barrier layer thickness, we prepared another group of MTJ devices with different GaSe thickness ranging from 6-layers to 11-layers. The optical images of devices A1, B1, C1, D1 and E1 are shown in **Figures R3a-e** respectively. The TMR- B curves of the devices measured at 10 K and 10 mV bias are shown in **Figures R3f-j** respectively. The extracted TMR value is shown in **Figures R3l**. With the increasing of the GaSe thickness, the TMR value first increases, then decreases, and the maximum value appears at 9-layer GaSe. As shown in **Figures R3k**, the J - V_{bias} curves at parallel state for different devices are still nonlinear and dispersed, indicating the transport is determined by the tunneling mechanism. In summary, the reproducibility of the barrier-thickness-dependent TMR characteristics has been proved. We add the related information in the Supplementary Information (Figure S6).

Figure R3 (a-e) The optical images of devices A1, B1, C1, D1, and E1, respectively. Scale bar: 10 μm. (f-j) The TMR- B curves of the devices measured at 10 K and 10 mV bias. (k) The J - V_{bias} curves at parallel state for the different devices. The inset shows the J - V_{bias} curve of the device A in a large bias range. (l) The extracted TMR value for the devices. The temperature is fixed at 10 K.

4. The authors claim that additional sign changes of TMRs can be related to the spin-polarized electronic structures of FGTs (Fig. 3a and 3b), and the multiple TMR sign changes are reproduced with spin-dependent tunneling models with considerations of chemical potentials and spin-dependent density of states (DOS) of FGTs. To be physically acceptable for applying the tunneling model, however, tunnel-barrier heights for the MTJs should be much higher than the chemical potentials of FGT-source and FGT-drain electrodes (Fig. 3b), allowing the electron tunnelings to be governed by the direct tunneling mechanism, just like the FGT-hBN-FGT device in the Ref 31. However, FGT-GaSe-FGT devices have much lower tunnel barrier heights (< 0.83 eV), which raises serious doubts about applying the tunnel model to explain the GaSe-layer-number-dependent TMR features, not to mention the ambiguities of explaining how the number of TMR-sign reversals are linked to the thickness of GaSe spacer.

Response: We thank the Referee for this insightful suggestion. As shown in **Figure R2**, the tunnel-barrier height is ~ 0.9 eV for FGT-GaSe-FGT device A-G ($t_{\text{GaSe}}=5.5$ -15.6 nm), which is not sensitive to the barrier layer thickness. In some previous researches, even the insulating

barrier were used, the barrier heights of CrI₃/h-BN, Fe/MgO and CoFe/MgO interfaces are estimated to be 0.25 eV [Z. Wang *et al. Nat. Commun.* 9, 2516 (2018)], 0.39 eV [S. Yuasa, *Nat. Mater.* 3, 868 (2004)], and 0.9 eV [J. S. Moodera and L. R. Kinder, *J Appl. Phys.* 79, 4724 (1996)], respectively. The lower barrier height of MgO-based MTJs could be attributed to the defect states in MgO barriers. With the barrier height for such systems is close to or even lower than that of our FGT-GaSe interface, they still show a perfect tunneling behavior. Therefore, the tunnel-barrier heights in FGT-GaSe-FGT device could be enough to allow the electron tunneling. Furthermore, the oscillation effect and sign-change behavior of the bias-dependent tunneling magnetoresistance could be attributed to the coherent tunneling [K. Dolui *et al. Phys. Rev. B* 90, 041401 (2014); Tsymbal, E. Y. & Žutić, I. *Handbook of Spin Transport and Magnetism Ch. 13 (CRC Press, 2019)*]. We add the related information to the main manuscript.

Reviewer #2 (Remarks to the Author):

The manuscript by Zhu et al. entitled « Large and tunable magnetoresistance in van der Waals Ferromagnet/Semiconductor junctions» addresses the topic of magnetic tunnel junctions with 2D-SC and FGT electrodes. The article shows very interesting results of MR for a system with two FGT electrodes and a GaSe tunnel barrier.

I overall find the results very interesting and encouraging. The observation of high MR at low temperature in such structure is inspiring for the community, opening new opportunities. With the following few points being clarified (notably the differences with ref Nat. Mater. 21,1144 (2022) already cited in the manuscript) this article will be a great contribution to the field and I would gladly recommend its publication in Nature Communications.

Here are my comments/questions to the authors:

1) Line 28-41 Overall the abstract and its claims could win from clarification. In the abstract, the authors say that the “Integration of two-dimensional ferromagnets in semiconductor-based vdW junctions offers gate-tunability, bias dependence, magnetic proximity effects, and spin-dependent optical-selection rules.” This sentence could be clarified as it is not clear what property is expected to come from the ferromagnet and what is expected to come from the semiconductor. Typically, the semiconductors can’t be gated within the junction and probably optical selection rules apply to the semiconductor... Also, the author should split the sentence “We demonstrate that not just the magnitude, but also the TMR sign is tuned by the applied bias or the semiconductor thickness, enabling modulation of highly spin-polarized carriers in vdW semiconductors.” as the TMR sign is not tuned by the thickness but by the applied bias while going to higher voltages. The authors should also specify that samples are based on flakes.

Response: We are grateful for such a positive assessment of our work and these valuable suggestions. One sentence “Integration of two-dimensional ferromagnets in semiconductor-based...” was revised to “Integration of semiconductors into the MTJs offers energy-band-tunability, bias dependence, magnetic proximity effects, and spin-dependent optical-selection rules”. We hope that this change helps to emphasize the role of semiconductors, including their spin-dependent optical-selection rules.

And the other sentence “We demonstrate that not just the magnitude, but also the TMR sign ...” was revised to “We demonstrate that not only the magnitude of the TMR is tuned by the semiconductor thickness but also the TMR sign can be reversed by varying the bias voltages,

enabling modulation of highly spin-polarized carriers in vdW semiconductors.”

In the caption of the newly added Fig. 1a, we clarify that the device is made of different flakes, which is also visible from provided optical image.

2) Line 49 In the sentence “MTJs with both conventional or vdW ferromagnets typically include an insulating spacer layer, which makes it easier to achieve high TMR compared to a semiconducting barrier.” Could the authors comment on why insulators should intrinsically give a lower MR than SC? As it seems it is not the presence of the insulator per se that gives high MR but more the specific filtering mechanism linked to certain ones such as MgO.

Response: Thank you for the Reviewer’s comments. According to previous papers, the TMR value of MTJ based on MgO and hBN insulators is often higher than that of MTJ devices based on semiconductor GaAs or MoS₂ [*H. William, Sci. Technol. Adv. Mater. 9, 014106 (2008)*]. Such a high TMR is beneficial to the spin-filtering effect of MgO and hBN. When the barrier layer is thin enough (< 5 nm), larger bandgap generally enhanced the effect of symmetry-derived spin-filtering (and hence the TMR) because the decay rates in the barrier are generally increased when the bandgap becomes larger. In semiconductors with smaller bandgaps, the spin-filtering effect can be enhanced by selecting the ferromagnetic electrode that matches its energy band, and by increasing the thickness of the semiconductor barrier [*L. Zhang et al. J. Phys. Chem. C 124, 27429 (2020)*; *K. Dolui et al. Phys. Rev. B 90, 041401 (2014)*]. In the revised main manuscript, the sentence “MTJs with both conventional or vdW ferromagnets typically include an insulating spacer layer, which makes it easier to achieve high TMR compared to a semiconducting barrier” was revised to “MTJs with both conventional or vdW ferromagnets typically include an insulating spacer layer instead of semiconducting barrier layer, owing to an extensive research on insulators such as Al₂O₃, MgO, and hBN”.

3) Line 55-58 Here if graphene is included in the discussion the work on 2DvdW could also include a more broaden literature (such as *Phys. Rev. B 90, 085429 (2014)*, *Sci. Rep.*6, 21168 (2016), *2D Mater.* 4 1014001 (2017), *ACS Nano*, 12, 4712 (2018), *2D Mater.* 7, 015026 (2020), *Nat Commun* 11, 3657 (2020), *Nat. Commun.* 11 5670 (2020), *Commun. Phys.* 4, 124 (2021)...). Line 64, reference should be made to *Nat. Mater.* 21,1144 (2022) for which 110% was found with WSe₂.

Response: We thank the Reviewer for these helpful improvements. The sentence “expanding the tunability of spin-dependent properties, as demonstrated, for example, through gate-tunable magnetic proximity effects in Co/graphene lateral spin valves” was revised to “expanding the

tunability of spin-dependent properties, as demonstrated, for example, through barrier-thickness controlled spin polarization and gate-tunable magnetic proximity effects in hybridized 2D material/ferromagnet interface for spin valves, and gate-tunable spin galvanic effect in van der Waals heterostructures of graphene with a semimetal or topological insulator.” The manuscript is modified as “Recently, the all-vdWs Fe₃GeTe₂(FGT)-based MTJs have been widely studied, among which a large TMR of ~300% (4.2 K), ~50% (10 K) and ~110% (4.2 K) have been observed in devices with insulating spacer hBN and devices with semiconductor InSe and WSe₂, respectively”. The reference of [K-H Min *et al. Nat. Mater.* 21,1144 (2022)] was cited. All of references mentioned in the comments are cited. We have further inserted another related work [P. Lazić *et al. Phys. Rev. B* 93, 241401(R) (2016)].

4) Line 85 Fig 1a. A picture of the real structure is missing. Could the authors include in the core text an actual image of the device where the different layers can be clearly distinguished alike Figure S1 (only figure S1 is too low resolution).

Response: To address this valuable suggestion, we have provided the optical image of the core structure of the device in the revised manuscript. The typical optical image of the core structure of the MTJ devices is plotted in **Fig. 1a**, where the two FGT electrodes sandwich a GaSe layer. To avoid oxidation, we also covered the core structure of the device with a hBN flake. The schematic diagram of the device and magnetotransport setup is shown in the inset of **Fig. 1a**, where an out-of-plane magnetic field **B** controls the magnetization alignment of the FGT electrodes. In addition, as shown in the **Figure R4** below, we also modified Figure S1 to make the optical image have a higher resolution.

Fig. 1 | Large TMR in the FGT/GaSe/FGT MTJ devices. **a**, Typical optical image of the core structure of the device made of different flakes. Inset shows the schematic diagram of the device and magnetotransport setup, where the magnetic field (B) is applied in out-of-plane direction.

Figure R4. (a-f) The optical images of the devices A-G, respectively. The insets of the figures (a-f) show the scanning thickness of GaSe flakes for devices A-G by AFM measurements. The FGT and GaSe flakes are outlined with red and orange dotted lines respectively. The scale bar is 10 μm .

5) Line 72 While the text gives a certain amount of details on the GaSe layers, information on the FGT layers are missing: like the thicknesses of both FGT layers in the devices.

Response: We appreciate this opportunity to provide the needed information. To have different coercive fields for the top and bottom electrodes, we select different thicknesses of FGT flakes [W. Zhu et al. *Adv. Mater.* 33, 2104658 (2021); C. Hu et al. *Sci. Bull.* 65, 1072 (2020); H. Lin et al. *ACS Appl. Mater. Inter.* 12, 43921 (2020)]. The thickness of the bottom and top FGT is ~ 8 nm and ~ 12 nm respectively, and the thickness of the middle GaSe ranges from 5.5 to 16.5 nm. We added the above information to the main manuscript.

6) Line 79 Fig 1b. The J-V curve for devices C and D which are the ones showing the highest MR do not really look like conventional non-linear tunneling curves (parabolic G-V) with some kind of kink at low voltage. Could the authors comment on that?

Response: We thank the Reviewer for this valuable comment. To understand the bias dependence of the tunneling current density, we analyzed two possible tunneling mechanisms by considering the barrier properties: (i) direct tunneling and (ii) F-N tunneling. When the

applied bias voltage is far less than the effective tunnel-barrier height ($V_{\text{bias}} \ll \phi_B$), the conduction current only has direct tunnel current, and the current density-voltage dependence of direct tunneling can be expressed by the Simmons' formula [*J. G. Simmons, J. Appl. Phys.* 34, 1793 (1963)]:

$$J = \frac{3q^2(2m^*q\phi_B)^{1/2}V_{\text{bias}}}{2h^2d} \exp\left[-\frac{4\pi d(2m^*q\phi_B)^{1/2}}{h}\right],$$

where q is the elementary charge, d is the barrier width, m^* is the effective electron mass, ϕ_B is the effective tunnel-barrier height and h is Planck's constant. Therefore, J should be a nearly linear function of V_{bias} . The current of devices C and D should be dominated by direct tunneling mechanism under small bias, so they have approximately linear J - V curve. In addition, the nonlinearity observed in the J - V curve is not obvious, while the obvious parabolic behavior can be observed in the R - V curve (as shown in the **Figure R5** below), and the change trend of resistance of different devices is very similar, indicating that they are tunneling devices. We add the related information in the supplementary information (Figure S3).

Figure R5. The resistances of the devices change with the bias voltage. In different devices, the change trend of the resistance with the bias voltage is almost the same. Under a large bias voltage, the resistances of the devices show parabolic behavior, indicating that they are all tunneling devices.

7) Line 114. The authors suggest that the MR can be improved by optimizing the thickness but this seems counterintuitive when looking at Fig1d. There, 200% seems to be the maximum. Also, Line 110 what exact mechanism do the authors involve to explain the reduction at low barrier thickness?

Response: We thank the Reviewer for an opportunity to clarify this situation and apologize for any misunderstanding. In the manuscript, we want to say that the TMR can be enhanced by optimizing the thickness of the barrier layer for all the semiconductor-based MTJs, not only for GaSe. The sentence is revised to “This experimental result suggests TMR in the semiconductor-based MTJs is directly related to the barrier layer thickness, which can be improved by optimizing the thickness and quality of the barrier layer”.

In line 110, the TMR is reduced at low-barrier thickness, which is attributed to the spin-filtering effect of GaSe that decreases with the decreasing thickness of GaSe barrier layer. Similar phenomena have been reported in some theoretical and experimental papers [*M. Covington et al. Appl. Phys. Lett. 76, 3965 (2000)*; *X. Jiang et al. Appl. Phys. Lett. 83, 5244 (2003)*; *K. Dolui et al. Phys. Rev. B 90, 041401 (2014)*].

8) Line 153 Fig3b appears puzzling to me and could win to be clarified. Why are the authors considering that P state carriers correspond to the lowest energy (E_f -drain) and AP state to the highest (E_f -source)? According to the picture given by the authors, carriers are tunneling from the right electrode to the left. More importantly the ones with lower tunnel barrier height, hence higher energy (at E_f -source), are the one that contribute the most to the total current. I would thus mainly expect carriers corresponding to the green arrows to dominate the current. What are the blue ones corresponding to?

Importantly, Fig2b and Fig3c and Fig4 look a lot like figures from *Nat. Mater.* 21,1144 (2022). Reference/comparison to this work should be mentioned in the discussion.

Response: We appreciate the Reviewers valuable comments and apologize for the misunderstanding caused by the arrows in the Fig3b. In the original Fig3b, both the lower blue arrows and the higher green arrows indicate that the electron tunneling from one spin band to another, rather than that the tunneling occurs only at the place marked by the arrows. To present this situation more clearly, we modified the Fig3b. As shown in the new Fig3b, in the parallel state, the minority spin bands of the left electrode tunneling to the minority spin bands of the right electrode, and the majority spin bands of the left electrode tunneling to the majority spin bands of the right electrode. As shown in Fig3c, in the antiparallel state, the minority spin bands of the left electrode tunneling to the majority spin bands of the right electrode, and the majority spin bands in left electrode tunneling to the minority spin bands of the right electrode. To simplify the calculation, we assume that the tunneling is elastic and spin-conserving, with the corresponding tunneling probability independent of the initial and final states as well as the

tunnel-barrier height for electrons.

In Fig2a-b, we study TMR vs V_{bias} curves for the devices with different GaSe layer thickness, and sign reversal of TMR is found in all devices, which is similar to results reported in Nat. Mater. 21, 1144 (2022). However, the critical bias voltage is different in our devices, when the TMR sign changes from positive to negative, such as 0.76 V for device B and 0.56 V for device D.

In Fig3c, by using a simplified elastic tunneling model, we can still get similar results as reported in Nat. Mater. 21, 1144 (2022), which is cited as Ref. 27.

In Fig4, we have studied the relationship between TMR and temperature, and proved that the TMR value decreases monotonously with the increase of temperature, which could be attributed to the reduction of spin polarization caused by the reduction of perpendicular anisotropic energy and the enhancement of thermal energy with temperature. However, in the Ref. 27, they did not study the temperature dependence of TMR.

Fig. 3 | The simulation of spin-resolved DOS of FGT and bias-dependent TMR calculated by elastic tunneling model. a, The calculated spin-resolved DOS for both the up (blue line) and down (red line) spins of the FGT in 3-layer-FGT/6-layer-GaSe/3-layer-FGT heterojunction.

b-c, Schematic diagram of direct band-to-band spin-dependent tunneling under bias window V_{bias} in the parallel and antiparallel states. Blue and red represent the up and down spin channels respectively. **d**, The calculated TMR as a function of V_{bias} by using the simple elastic tunneling formula.

9) Line 180 As core to the paper, the model could benefit from further discussion and explanation of the physical mechanism at play.

Also, it was suggested in Nat. Mater. 21,1144 (2022) that the reversal of MR as a function of bias was occurring for h-BN, WSe2 and WS2. The authors should compare their result with this work.

Response: We thank the Reviewer for these suggestions. The Nat. Mater. 21,1144 (2022) has studied TMR properties of ferromagnetic junctions, and tests the TMR vs V_{bias} curves of the devices with different barrier layer thickness. They observed that the sign reversal behavior of TMR with the increase of bias voltage in all the devices. And this complex reversal behavior of TMR is explained by invoking exclusively the density of state of the FGT electrode rather than the parameters of the tunnel barrier. Their results show that the properties of TMR are independent of the barrier layer thickness.

In our manuscript, in addition to verifying the sign reversal of TMR with increasing bias voltage, we also studied the effect of barrier layer thickness on the TMR properties.

On the one hand, we found that with the increase of the barrier layer thickness, there is a maximum value of TMR at a certain thickness. This should be due to the enhancement of spin-filtering effect with the increase of barrier layer thickness, while with the further increase of the barrier layer thickness, the effect of impurity scattering in the barrier on the tunneling electrons will be enhanced, resulting in the rapid reduction of TMR.

On the other hand, we also found that the oscillation period of TMR under applied high bias voltage is related to the barrier layer thickness. As the barrier layer thickness increases, the oscillation period decreases, which is caused by the coherent tunneling [*K. Dolui et al. Phys. Rev. B 90, 041401 (2014)*; *Tsybal, E. Y. & Žutić, I. Handbook of Spin Transport and Magnetism Ch. 13 (CRC Press, 2019)*].

10) Line 186 An experimental demonstration involving indirect band gap with states away from the gamma point in 2D SC was previously discussed (ACS Nano 13, 14468 (2019)). Line 190 Have the authors been calculating such matching? If yes, could they provide some feedback.

Response: We thank the Reviewer for the opportunity to clarify this issue. We quoted [ACS

Nano 13, 14468 (2019)] for reference. **Figure R6** shows the majority and minority spins Fermi surfaces of three-layer Fe_3GeTe_2 at different energies ranging from $E_F-0.2$ to $E_F+0.2$ eV, where “0” indicates the Fermi energy. Majority spins have multiple bands at the Fermi energy covering the large portion of the 2D Brillouin zone (BZ), the minority spins have only a few states available, resulting in a large area of the 2D BZ with no overlap. Based on the Julliere model, this will give a large positive magnetoresistance. When voltage is applied to the MTJ device, the energy surface of one of the Fe_3GeTe_2 electrodes will move relatively to the other electrode, which makes the majority spins (or minority spins) of one electrode overlap with the minority spins (or majority spins) of the other electrode. As shown in **Figure R6**, in case of $E_F+0.2$ eV, some bands of majority and minority spins overlaps. As a result, the negative magnetoresistance will appear. We add the related information in the Supplementary Information (Figure S13).

Figure R6. Majority and minority spins Fermi surfaces of three-layer Fe_3GeTe_2 at different energies ranging from $E_F-0.2$ to $E_F+0.2$ eV, where “0” indicates the Fermi energy. Colors indicate the Fermi surfaces belonging to different bands.

11) Line 217 and 226 the authors demonstrate a spin polarization of 70%, it is worth to compare it with other results on other 2D SC based MTJs with different barriers such as Nat. Mater. 21,1144 (2022) and even other 2Ds such as the -98% observed for graphene ACS Nano 16, 14007 (2022).

Response: We thank the Reviewer for bringing to our attention this additional work and the opportunity to put our findings in a broader context. We quoted the spin polarizations in these two papers. The manuscript is modified as “which is larger than that obtained in other 2D

semiconductor-based MTJs, such as 45% in Fe₃GeTe₂/InSe and 60% in Fe₃GeTe₂/WSe₂ [*W. Zhu et al. Adv. Mater.* 33, 2104658 (2021); *K. H. Min et al. Nat. Mater.* 21, 1144 (2022)], but lower than that of Ni(111)/Gr interface”.

12) Fig S2 raises some questions. Why can one observe a jump in FigS2c for qPhy, whereas it is not seen at all in Fig1c, in either the TMR or R.A vs thickness plots.

Response: We thank the Reviewer for this important question. This jump in Fig.S2c is caused by different tunneling mechanisms in different devices. In devices E and F and G, the direct tunneling current is very small due to the larger barrier thickness. Under a large applied bias voltage, F-N tunneling current dominates, so the effective barrier height obtained by fitting the F-N tunneling formula is more reliable. In devices C and D, due to the relatively small barrier thickness, the direct tunneling current is large and cannot be ignored. At this time, under a large applied bias voltage, the conductive current is determined by both the direct tunneling and the F-N tunneling currents, so the effective tunnel-barrier height obtained by fitting the F-N tunneling formula, has deviation and is no longer strictly valid. In addition, under a small bias voltage, we can get the effective tunnel-barrier height by fitting the direct tunneling formula. For a specific theoretical calculation, please refer to Figure R2 and corresponding response to Reviewer #1.

Reviewers' Comments:

Reviewer #1:

Remarks to the Author:

Comments for Authors:

In the revised manuscript, the authors have added a new data set of FGT/GaSe/FGT spin valves with varying GaSe thickness and reproduced their initial findings of layer-number dependent TMR variations. Additionally, the authors have revised the manuscript to better present the potential role of the semiconducting GaSe tunnel barrier in low-dimensional spintronic applications. However, while the authors' claim that spin-filtering effects can be attributed to the observed thickness-dependent TMR variations, I am not fully convinced by the authors' claim since no conclusive theoretical and experimental works supporting that spins can be effectively manipulated in the semiconducting GaSe layers are presented in the current manuscript, other than citing previous reports. Despite this, I believe that the manuscript has a high potential to draw wide interest from the readers in Nature Communications, and prompt publication can be beneficial for spintronic communities. Thus, I would gladly recommend its publication in Nature Communications.

A quick question on the thickness-dependent oscillation period of TMRs under high bias is the following. The authors have listed coherent electron tunneling through the GaSe tunnel barriers as the main attribute for the TMR sign changes. I agree with the authors that multiple TMR sign changes observed at the device E can be explained by the elastic tunneling model, as elaborated in detail in Figure 3. As displayed in Figure 3d, the multiple TMR sign changes are expected in the bias window of -2 V and 2 V, which is consistent with the TMR variations from device E (Figure 4c). However, for the differing V_b -dependent TMR characteristics observed in the devices B and D with thinner GaSe tunnel barriers (Figures 4a and 4b), the authors have relied on a rather ambiguous argument of how much coherent electron tunneling is dominant in the vertical spin flows.

My impression is that the sample-to-sample variation could be closely related to the tunnel junction resistance values. With semiconducting tunnel barriers as thick as the GaSe film in device E, the elastic tunneling model with FGT electronic structures can be applicable since all the voltage drops of the device should be happening at the tunnel junctions. With thinner semiconducting films and lower tunnel junction resistances, however, the bias voltage effectively applied to the tunnel junctions should be smaller than the voltage driven by an external voltage source: this phenomenon is often dubbed as a voltage-divider effect. Based on this, bias voltages where the TMR-sign reversals occur should be larger in devices with thinner tunnel barriers, thus lower tunnel-junction resistances, as shown in Figure 2d for devices B, D and E.

Reviewer #2:

None

Reviewer #1:

Comments to the authors:

In the revised manuscript, the authors have added a new data set of FGT/GaSe/FGT spin valves with varying GaSe thickness and reproduced their initial findings of layer-number dependent TMR variations. Additionally, the authors have revised the manuscript to better present the potential role of the semiconducting GaSe tunnel barrier in low-dimensional spintronic applications. However, while the authors' claim that spin-filtering effects can be attributed to the observed thickness-dependent TMR variations, I am not fully convinced by the authors' claim since no conclusive theoretical and experimental works supporting that spins can be effectively manipulated in the semiconducting GaSe layers are presented in the current manuscript, other than citing previous reports. Despite this, I believe that the manuscript has a high potential to draw wide interest from the readers in Nature Communications, and prompt publication can be beneficial for spintronic communities. Thus, I would gladly recommend its publication in Nature Communications.

Response:

We sincerely thank the reviewer for his/her very positive feedback and recommendation the prompt publication of our paper in Nature Communications. As for the comment regarding spin-filtering effects in the semiconducting GaSe layers, the reviewer has asked it specifically in the quick question, we've fully addressed this in the response to his/her quick question as shown below.

A quick question on the thickness-dependent oscillation period of TMRs under high bias is the following. The authors have listed coherent electron tunneling through the GaSe tunnel barriers as the main attribute for the TMR sign changes. I agree with the authors that multiple TMR sign changes observed at the device E can be explained by the elastic tunneling model, as elaborated in detail in Figure 3. As displayed in Figure 3d, the multiple TMR sign changes are expected in the bias window of -2 V and 2 V, which is consistent with the TMR variations from device E (Figure 4c). However, for the differing V_b -dependent TMR characteristics observed in the devices B and D with thinner GaSe tunnel barriers (Figures 4a and 4b), the authors have relied on a rather ambiguous argument of how much coherent electron tunneling is dominant in the vertical spin flows.

My impression is that the sample-to-sample variation could be closely related to the tunnel junction resistance values. With semiconducting tunnel barriers as thick as the GaSe film in device E, the elastic tunneling model with FGT electronic structures can be applicable since all the voltage drops of the device should be happening at the tunnel junctions. With thinner semiconducting films and lower tunnel junction resistances, however, the bias voltage effectively applied to the tunnel junctions should be smaller than the voltage driven by an external voltage source: this phenomenon is often dubbed as a voltage-divider effect. Based on this, bias voltages where the TMR-sign reversals occur should be larger in devices with thinner tunnel barriers, thus lower tunnel-junction resistances, as shown in Figure 2d for devices B, D and E.

Response: We thank the Referee for this insightful suggestion. To explain the barrier thickness-dependent TMR variation originating from the spin filtering effect of the barrier layer GaSe, we calculated the TMRs of the variable barrier layer thickness using the first principles calculation method, the device configuration is shown in the following Fig. R1a. As the thickness of the GaSe layer increases from monolayer to tri-layer, the TMR of the device first increases and then decreases, showing a similar variation pattern to the experimental results (Fig. R1b). Our theoretical calculation shows a maximum TMR of 24490% in bilayer GaSe device, while the experimental observation shows a maximum TMR of 192% in eight layers of GaSe devices. The reason why the calculated TMR is much greater than the experimental observation value may be attributed to unfavorable factors such as interface scattering, defect scattering, and pinhole effect in the experimental device. The reason for the difference in GaSe thickness corresponding to the maximum TMR may be that the interface roughness of the GaSe barrier layer in the experimental device results in the actual barrier layer thickness being smaller than the AFM measurement value. Besides, the spin polarization of Fe_3GeTe_2 at the Fermi level is only 43%, and the TMR calculated by the Julliere model without considering the spin filtering effect of the barrier layer is only 45% (Fig. 3a,3d in the main manuscript), while the TMRs calculated theoretically and observed experimentally is far more than 45%, which verifies the existence of the spin filtering effect of the barrier layer.

(a) Device configuration:

Cu electrode Fe₃GeTe₂ (2L) GaSe (nL) Fe₃GeTe₂ (2L) Cu electrode

(b)

Figure R1. (a) The structure of the device used for theoretical calculation of TMR. (b) The calculated thickness-dependent TMR as the GaSe barrier layer changing from monolayer to tri-layer.

Then, accordingly to the Referee's valuable suggestion, we extracted the bias voltages of the first TMR sign reversal at 10 K as a function of device's resistance, as shown in the **Figure R2** below, the voltage values increases as the device resistance (or tunnel barrier thickness) decreases, indicating voltage-divider effect could also be the reason for an increase in oscillation period in thinner barrier layer devices. We have revised the main manuscript accordingly, and added **Figure R2** as **Figure S14** in the Supplementary Information, and marked in red.

Figure R2. Bias voltage of the first TMR sign reversal at 10 K as a function of device resistance.

Reviewers' Comments:

Reviewer #1:

Remarks to the Author:

I appreciate the authors' efforts in addressing my previous concerns in the revised manuscript. Based on this, I recommend the publication of the manuscript in Nature Communications.

Reviewer #1 (Remarks to the Author):

I appreciate the authors' efforts in addressing my previous concerns in the revised manuscript. Based on this, I recommend the publication of the manuscript in Nature Communications.

Response:

We sincerely thank the reviewer for recommending our paper for publication in Nature Communications.